# Self-Healing and Highly Stretchable Hydrogel for Interfacial Compatible Flexible Paper-Based Micro-Supercapacitor

**DOI:** 10.3390/ma14081852

**Published:** 2021-04-08

**Authors:** Yutian Wang, Yunhui Shi, Yifan Gu, Pan Xue, Xinhua Xu

**Affiliations:** 1School of Materials Science and Engineering, Tianjin University, Tianjin 300072, China; 15022345024@163.com (Y.W.); yhshi@hebut.edu.cn (Y.S.); gyf18822066907@163.com (Y.G.); xp784422160@163.com (P.X.); 2Tianjin Key Laboratory of Composite and Functional Materials, Tianjin University, Tianjin 300072, China

**Keywords:** hydrogel electrolytes, stretchability, dual-dynamic, self-healing, flexible micro-supercapacitors

## Abstract

Most reported wearable electronic devices lack self-healing chemistry and flexible function to maintain stable energy output while irreversible damages and complex deformations. In this work, we report a dual-dynamic network electrolyte synthesized by micellar elastomers introduced into strong hydrogel matrix. The gel electrolyte is fabricated by physically cross-linking the borax-polyvinyl alcohol (B-PVA) network as tough matrix and poly (ethylene oxide) (PEO)-poly (propylene oxide) (PPO)-poly (ethylene oxide) (Pluronic) to frame elastic network, followed by immersion in potassium chloride solution. Under the action of dynamic borate ester bond and multi-network hydrogen bond, the as-prepared electrolyte exhibits high stretchability (1535%) and good self-healing efficiency. Based on the electrolyte, we assemble the interfacial compatible micro-supercapacitor (MSC) by multi-walled carbon nanotubes (MWCNT) interdigital electrode printed on cellulosic paper by direct ink writing (DIW) technique. Thanks to the large specific area and compressive deformation resistance of cellulosic paper, the MSC with tightly interfacial contact achieves high volumetric capacitance of 801.9 mF cm^−3^ at the current density of 20 μA cm^−2^. In the absence of stimulation of the external environment, the self-healing MSC demonstrates an ideal capacity retention (90.43%) after five physical damaged/healing cycles. Our research provides a clean and effective strategy to construct wearable MSC.

## 1. Introduction

Recently, research has showcased vigorous progress in developing a variety of wearable electronic products including portable biosensor, flexible electrical display and smart watch [1,2,3]. However, the demonstration of flexible power sources lags behind. To power such late-model electronics, the current energy storage systems need to be redesigned. Compared to batteries, supercapacitors possess achieving long life cycles, high power density and efficient charging/discharging speed showing application prospect in wearable electronics [4]. Conventionally, supercapacitors with traditional structure suffer from twisting, bending or complex deformations, leading to irreversible structural damage and performance degradation. Such faults seriously restrict the applicability and reliability [5]. Meanwhile, the heavy weight of conventional supercapacitors limits the flexibility of electronics [6,7]. Thus, requirements for portable applications call for miniaturization, mechanical stability, flexibility and interfacial adhesion capability to construct flexibility electronics.

Self-healing materials are the key to cope with internal or external damage. Over the last decades, a lot of studies have focused on the field of the elastic recovery, structural restoration and functional self-healing [8,9]. In a variety of self-healable materials, the self-healing hydrogels realize ion transport while maintaining outstanding mechanical stability showing great potential as a flexible electrolyte [10]. An intrinsic self-healable hydrogel repairs damage through constructing dynamic interactions in polymer network, improving the service and reliability of hydrogel [11]. According to the self-healing mechanism, the self-healing hydrogels are classified as dynamic noncovalent bond type. The dynamic noncovalent bond contains chain entanglement [12], hydrogen bonding [13] and supramolecular interaction [14], whereas the dynamic covalent bond involves phenyl borate [15], disulfide bond [16], and hydrazide bond [17]. As a well-known self-healing polymer, polyvinyl alcohol (PVA) has advantages on low cost, easy processability, good biocompatibility and biocompatibility [18]. Due to the plenty of hydroxyl functional groups on the molecular chain skeleton, PVA hydrogel can self-heal spontaneously with hydrogen bonding between molecular chains and new interactions (dynamic borate bond, coordination bond and hydrogen bond) [3,12]. It is worth pointing out the inherent defects of PVA based hydrogels, such as long self-healing cycle and poor stretchability restrict the practical application [19]. Pluronic triblock linear copolymer chains agglomerate into micelles irreversibly at the increased temperature, with hydrophilic shell formed by poly (ethylene oxide) (PEO) blocks and hydrophobic micellar core formed by poly (propylene oxide) (PPO) blocks [20,21,22,23]. The existing research indicates that the Pluronic aqueous solution with high concentration can improve the viscoelastic modulus dramatically [24]. Therefore, it is of great importance to introduce Pluronic into PVA hydrogel to enhance self-healability, flexibility and stability.

Three-dimensional (3D) printing technique can manufacture the 3D physical consistent with the corresponding mathematical mode by stacking materials layer by layer [25,26,27]. Benefiting from its process flexibility and large-scale scalability, the cost-effective direct ink writing (DIW) method, known as 3D printing based on ink extrusion, opens the way for the next-generation power sources [28]. In particular, DIW technique is capable of manufacturing fine and complicated microstructures with high-precise and personalized customization, meeting the requirements of miniaturized energy storage devices [29]. Micro-supercapacitors (MSC)s are supposed to the state-of-the-art miniature power sources, characterized by quick charging/discharging rate, stable cycle behavior along with efficient ion transmission [30,31,32,33]. Developing multifunctional MSCs can promote the potential application in intelligent on-chip microelectronics [29]. Compared with plastic carrier or rigid substrate, cellulose paper possess flexibility, lightweight, and low cost [34]. Conductive paper-based devices, as wrapping cellulose paper with conductive layer, is widely applied in nursing point test [35], biological fuel cell [36] and light-emitting diodes [37]. Adopting cellulose paper as the substrate, late-model flexible MSCs are expected to undergo various forms of deformation without interlayer separation, adapting to the complex working environment of flexible wearable devices.

In this work, the novel dual-network gel electrolyte is fabricated by immersing hydrogen bonded crosslinked borax-polyvinyl alcohol (B-PVA) and Pluronic into potassium chloride solution. B-PVA as the first network, based on the dynamic borate ester bond, provides strength and toughness for the hydrogel. Pluronic as the second network playing the role of plasticizer and elastomer enhances flexibility and mechanical elasticity of hydrogel. Compared with traditional PVA based hydrogel, Pluronic triblock polymer results in the micellization, microphase separation and physical cross-linking via the self-assembly of micelles, thus promoting the efficiency of self-healing behavior [38]. Optimizing preparing strategy of planar micro electrodes, the MSC can be assembled by attaching the electrolyte tightly to the multi-walled carbon nanotubes (MWCNT) interdigital electrode fabricated by printing MWCNT ink onto the cellulosic paper by DIW technique. Owing to the large surface area and flexibility of cellulose paper [34], the electrode layer without binder can have close interface contacted with substrate, in favor of improving electronic transfer capability.

## 2. Materials and Methods

### 2.1. Chemicals and Materials

PVA (degree of hydrolysis 99% and degrees of polymerization are 1700) was purchased from Meryer Chemical Technology (Shanghai, China). The Pluronic (Pluronic F127) (MW ca. 12,600) was purchased from Sigma-Aldrich (Shanghai, China). Borax (analysis of pure) was purchased from Tianjin Jiangtian Industrial Corporation (Tianjin, China). KCl (analysis of pure) was purchased from Aladdin Industrial Corporation (Shanghai, China). Carbon nanotube dispersion (10 wt% in water) was purchased from Chengdu Organic Chemistry Corporation (Chengdu, China). All the reagents without purification are directly used.

### 2.2. Synthesis of BPVA-Pluronic Gel Electrolyte

BPVA-Pluronic gel electrolyte was synthesized through a simple and fast process. 1.6 g PVA was first dissolved in 8.4 mL deionized water at 95 °C for 1 h and 0.075 g borax was dissolved in 5 mL deionized water at room temperature. Then 1.6 g Pluronic was added into the PVA solution with stirring at 60 °C for 2 h and two as-dissolved solutions were mixed together followed by stirring overnight at 60 °C. The resulting hydrogel was rinsed with deionization water to remove impurities and then equilibrated in 1 M KCl aqueous solutions for 4 h. The as-prepared elastomer is employed as BPVA-Pluronic-5 hydrogel.

With method mentioned above, more hydrogels were prepared by adjusting the proportion of Pluronic and PVA in the condition that the total mass fraction of polymer in hydrogels remained unchanged, named as BPVA-Pluronic-3 and BPVA-Pluronic-6. (Appendix A).

### 2.3. Fabrication of Flexible Electrochemical MSC

The DIW functions based on our former reported work [29,38]. The carbon nanotube dispersion was loaded into the plastic extrusion syringe as electrode ink, assembled with nozzle made of stainless steel (inner diameter of 340 μm). The extrusion rate of electrode ink was driven from G-code-controlled screws. The interdigitated structure preprogrammed by Solid Works was printed directly onto the paper substrate, controlled by multi-axial positioning system. The flexible MSC was subsequently dried in vacuum drying oven under the absolute pressure of 15.20 kPa at 60 °C overnight to evaporate the electrode solvent. The hydrogel was tailored to the required size and attached to the interdigital electrodes. After pressing the MSC under the glass slide overnight, the self-healing MSC was obtained.

### 2.4. Structural Characterization

The functional groups of the materials were investigated by The Fourier transforms infrared (Nicolet 360 spectrometer, FTIR) spectra (Thermo Fisher Scientific, Waltham, MA, USA). Solved by D_2_O, the chemical structures of the materials were evaluated by ^1^H nuclear magnetic resonance (Varian Inova 500 MHz, ^1^H NMR) spectra (Varian Medical Systems, Palo Alto, CA, USA). Energy dispersive X-ray spectroscopy (EDX) and micromorphology of hydrogels was researched by a scanning electron microscope (Hitachi S-4800 microscope, SEM) (Hitachi, Tokyo, Japan). The tensile samples were measured on a universal testing machine INSTRON 5965 (Instron, Boston, MA, USA) at varying drawing rates. The rheological behavior of carbon nanotube dispersion was analyzed by TA instruments-dhr-2 (Waters, Milford, MA, USA). 3D surface profile of the flexible MSC was studied by S neox Five Axis (Sensofar, Barcelona, Spain).

### 2.5. Electrochemical Characterization

The Electrochemical impedance spectroscopy (EIS) (Chenhua, Shanghai, China) test was conducted with an electrochemical working station (CHI660E). The as-prepared hydrogel electrolyte was pressed between two slides to form the polymer film. The hydrogel film was cut in appropriate size of Φ 16 mm × 1.28 mm and then attached between two stainless steel sheets of Φ 22 mm × 0.5 mm. Both sides of test sample were connected to positive and negative poles of electrochemical workstation respectively. The expression of the ionic conductivity was calculated on the basis of the following formulas:(1)$$\sigma \;=\;L/\left(RA\right)$$
where *L* is the thickness of the electrolyte, *R* is the bulk resistance and *A* is the cross-sectional area of the electrolyte.

The operating voltage window was determined by linear sweep voltammogram (LSV) test on the CHI660E station. Self-discharge test was measured on the CHI660E station and potential-specific capacity curve was tested on LAND battery test system CT2001A (Land, Wuhan, China). Next, Cyclic voltammetry (CV) and galvanostatic charge/discharge (GCD) measurements of the MSCs were tested in a potential window of 0 to 0.8 V on the CHI660E station. The capacitance (C_MSC_) of the MSC was obtained from the CV Curves in accordance with the following Equation (2):(2)$$C_{MSC}=it/V$$
where *i* is the current density, *t* is the time of discharging and *V* is the voltage window when discharging. The specific capacitance performance of the MSC was figured out by the following equations:(3)$$C_{areal}\;=\;C_{MSC}/A$$
(4)$$C_{volumetric}\;=\;C_{areal}/h$$
where *A* reveals the area of the MSC and *h* represents the thickness of interdigital electrode.

## 3. Results and Discussion

The preparation process of the BPVA-Pluronic hydrogel is shown in Figure 1a. A cross-linking network is formed between PVA and borax by dynamic borate bond. Owing to the reversibility of borate ester bond, the PVA hydrogel is restricted by weak mechanical behavior and insufficient stability, which limit application prospects of the hydrogel. Building the second network is an effective strategy to address the problems [3]. Here, we introduce Pluronic to construct the micellar network. Meanwhile, molecular chain entanglement and hydrogen bonds between molecular chains also form between B-PVA network and Pluronic micellar network [24]. To confirm the stability of BPVA-Pluronic hydrogel, two kinds of cuboid-shaped hydrogels are placed for 2 h at room temperature (Figure 1b). PVA hydrogel (the same preparation process without the addition of Pluronic) fails to maintain its original shape after 2 h, indicating that the bond in pure PVA hydrogel is dynamic. However, the BPVA-Pluronic-5 hydrogel remains unchanged as initial and no spill of solvent is observed, proving the enhancement of stability. This is mainly attributed to the formation of hydrogen bonds between molecular chains, reducing the density of intra PVA chain hydrogen bonds [12]. The better mechanical property provides wider application space for BPVA-Pluronic hydrogel.

The functional groups of PVA, Pluronic and BPVA-Pluronic-3 hydrogel were characterized by FTIR. As is shown in Figure 2a, 3308 cm^−1^ and 1342 cm^−1^ are assigned to the O-H stretching vibration peak and O-H in-plane bending vibration peak of alcohols [3]. In addition, the characteristic peaks in the chain of Pluronic are observed at 2871 cm^−1^ (C-H stretching vibration peak), 1373 cm^−1^ (C-H non-planar rocking vibration peak), 1100 cm^−1^ (C-O bending vibration peak) and 934 cm^−1^ (C-H rocking vibration peak) [23,24]. In comparison with the curve of pure PVA, O-H stretching vibration peak at 3308 cm^−1^ in the BPVA-Pluronic-3 hydrogel shifts to higher frequencies, proving the existence of hydrogen bonding between PVA molecular chains and Pluronic molecular chains [38]. On the meanwhile, ^1^H NMR spectra (Figure 2b–e) were used to further prove the chemical structures. The proton signals from PVA-segments detected at 1.56 and 3.89 ppm are ascribed to protons at the a and b positions, respectively. The proton signal of Pluronic segments at the c, d, e and f positions are also observed at 3.56, 3.46 and 1.02 ppm [18]. It is certain that the hydrogel electrolyte of the MSC has been synthesized successfully, and there are hydrogen bond interactions between the B-PVA network and Pluronic micellar network. With the increasing of Pluronic weight ratio, no obvious change is observed among the frequencies of characteristic peaks that belongs to the block polymer molecular chains (Figure 2f). Nevertheless, the distinguishing peaks of O-H stretching vibration and O-H in-plane bending vibration reduce with the decrease of PVA concentration, illustrating the lower degree of hydrogen bond density in the hydrogel.

To assess the interior micromorphology of hydrogel samples, the freeze-dried PVA and BPVA-Pluronic hydrogels were characterized by SEM. Figure 3a depict the section structure of pure PVA hydrogel. The PVA hydrogel exhibits the precipitation of KCl crystals on rough cross section. Figure 3b–d depict the section structure of BPVA-Pluronic-3 hydrogel. Compared with the interior morphologies of PVA hydrogel, all kinds of BPVA-Pluronic hydrogels (Figure 3c–h) possess a porous crosslinked network structure owing to the formation of the second network [14]. This type of structure can efficiently enhance the salt tolerance of double network hydrogel electrolytes and realize the creation of electro-conductive paths, thus promoting the transmission and diffusion of ions, which contributes to achieve high conductivity of the hydrogel electrolytes in theory [3]. By adjusting the ratio of Pluronic in polymer, there are differences in the interconnecting hole sizes and morphologies. The average void diameters ranged from tens of nanometers to 20 μm for the BPVA-Pluronic-3 hydrogel, ranged from tens of nanometers to 5.46 μm for the BPVA-Pluronic-5 hydrogel, ranged from tens of nanometers to 17.28 μm for the BPVA-Pluronic-6 hydrogel. The void size of the macroporous decreases first and then increases, attributing to the increase of pore density in hydrogel leading to the formation of new macropores. With increase of Pluronic content, the cross-section of hydrogel changes from a planar porous network structure with interlayer crosslinking to a 3D interconnected porous network structure at high magnification, possessing thinner network matrix. The results suggest that the introduction of Pluronic increases the density and path of the crosslinked network, expected to improve the elastic property and self-healing ability of PVA-based hydrogel [22,23]. To demonstrate the elemental composition and distribution of hydrogel, EDX spectrums are present in Appendix A. The polymer composed of C and O elements forms the matrix of the crosslinking network and B elements are distributed in the network skeleton as crosslinking sites. The distribution of K and Cl elements in the hydrogel is uniform, indicating the sufficient infiltration of salt solution.

To further study the influence of molecular chain network motion, the stress–strain behaviors of the BPVA-Pluronic-5 hydrogel were measured at various stretching speed. As Appendix A depicts, the stress-strain curves consist of elastic deformation zone, yield point and plastic deformation zone [38]. Strain increases significantly with strain in elastic deformation stage, the movement of molecular chain and the relative slip between chains lead to a neck extension in plastic deformation stage. With the increasing of the strain speed, the breaking tenacity of the hydrogel increases but its elongation at break decreases, which is a typical characteristic of elastomers [1,2]. At the stretching rate of 230 mm min^−1^, stress maintains a transient stay and then increases to the breaking stress. At the stretching rate of 100 mm min^−1^, the stress decreases with the increase of strain until the spline breaks. The stretchability of the BPVA-Pluronic-5 hydrogel also depends on the change of tensile rate. The maximum stretchability can exceed 4000% at the stretching rate of 100 mm min^−1^. This result can be contributed to the fact that more time is available for the re-formation of dynamic borate bond and the motion of polymer chain at low strain rate [18].

Mechanical properties of hydrogels (containing 20 wt% polymer materials) with varying proportions of PVA/Pluronic were estimated to explore the mechanism of Pluronic micellar network, as shown in Figure 4a. Compared with the pure PVA hydrogel, BPVA-Pluronic hydrogel overcomes the defects of low strength and cracking easily through the addition of second network, thus improving the elastic modulus and mechanical strength of the hydrogel. The fracture strength of the BPVA-Pluronic-3 hydrogel is 52.57 kPa, which is almost three times higher than that of the pure PVA hydrogel, and the fracture elongation is 416%, which is also 1.4 times as much as that of the pure PVA hydrogel. With the addition of Pluronic, the corresponding ruptured strain of the hydrogel raises. When the content of Pluronic is 2 (weight ratio to PVA), the hydrogel exhibits an excellent stretchability with the ruptured strain of 4866%, which is nearly 16.5 times longer than that of the PVA hydrogel without Pluronic, whereas slight decrease is observed between their fracture stress. The reduction of tensile fracture strength is owing to the decrease of hydrogen bond between molecular chains and the plasticizing mechanism of the Pluronic micellar network, leading to the decrease of crystallinity, being in agreement with the conclusion of FTIR (Figure 2f) [38,39].

To qualitatively study the self-healing behavior, the hydrogel block was cut in half with scissors and subsequently the two cross-sections were put together into contact at room temperature in absence of external environment stimulation. The self-healing behavior occurs at the interface of the sample initially after 2 min and the healed hydrogel can again bear a large shape change after 1 h (Figure 4c). For understanding the self-healing efficiency of BPVA-Pluronic-5 hydrogel in depth, the tensile test was conducted at different levels of healing time (Figure 4b). The change trend of stress-strain curve is consistent with the expectation that the longer healing time leads to the higher healed elongation at break. After 1 h of self-healing, the fracture strain of the healed hydrogel is 1480%, leading to the high healing efficiency (96.29%). Compared with traditional PVA hydrogels (Appendix A), the excellent self-healing property of hydrogel is mainly derived from the Pluronic micellar network, which promotes the occurrence of self-healing behavior [21,24]. On the fracture surface, Pluronic chains construct a new network through micellization and then drag the PVA chains through the hydrogen bond to induce the recombination of dynamic borate bond between broken hydrogels [18,39]. The PVA based hydrogel (BPVA-Pluronic-5) with both tensile property and self-healing efficiency is rare, beneficial to the development and utilization of flexible energy storage devices [31].

In addition to the requirements for mechanical tensile properties, high ionic conductivity and facile ionic transport are also important [5]. Appendix A shows the Nyquist plot of BPVA-Pluronic electrolyte with 1 weight ratio PVA/Pluronic containing different concentration of KCl. The ionic conductivity increases at low concentration and then decreases with the adding of KCl. With the KCl content of 1 mol/L, the BPVA-Pluronic electrolyte possesses high ionic conductivity of 27.57 ms cm^−1^. On the basis of equation *σ* = *nqμ*, the hydrogel conductivity is positive correlation with the mobility and carrier concentration [34]. And the mobility is inversely proportional to viscosity of the hydrogel according to *μ* = *q*/6*πrη* [33]. Therefore, the uniform distribution of KCl in 3D-network configuration of BPVA-Pluronic electrolyte constructs a transmission channel between KCl and the hydrogel, which increases the conductivity of electrolyte with the KCl addition [32]. However, due to the salt-out effect, the addition of the salt improves the viscosity of the electrolyte, thus reducing the ionic conductivity at high salt concentration [3]. The Nyquist plot of BPVA-Pluronic electrolytes with varied PVA/Pluronic ratio is shown in Appendix A. With the increase of Pluronic mass ratio, BPVA-Pluronic-5 electrolyte has better ionic conductivity than BPVA-Pluronic-3 electrolyte, originating from the synergistic effect of plasticizing effect and efficient stereoscopic micro crosslinking network. However, there is no obvious change between ionic conductivity of BPVA-Pluronic-5 and BPVA-Pluronic-6. This phenomenon mainly attributes to the reason that the larger void diameters and plasticizing effect of BPVA-Pluronic-6 counteract each other (Appendix A). The good ion transport of BPVA-Pluronic electrolyte is mainly attributed to the following reasons. (1) The plasticizing effect of the Pluronic elastomer results in the viscosity reduction of hydrogel. (2) 3D network structure provides an efficient ion transport channel. (3) Increasing the concentration of KCl can produce surplus of ions [6].

To explore the potential prospect of the hydrogel electrolyte in flexible energy storage device, self-healing MSCs were fabricated based on BPVA-Pluronic electrolyte film via DIW technology (Figure 5a) [29]. In consideration of features of the electrolyte and the flexible device, such as bending ability, self-healing capability and ionic conductivity BPVA-Pluronic electrolyte with 1 weight ratio Pluronic to PVA and 1 M KCl (BPVA-Pluronic-5) was chosen to improve the performance of smart MSCs. First, the carbon nanotube was printed onto the cellulosic paper acting as the flexible substrate [40,41,42,43]. As Appendix A shown, the rheological behavior of the carbon nanotube dispersion was estimated and the shear thinning phenomenon indicates that the ink is non-Newtonian fluid. At the plateau range from 0.1 to 1 Pa, the storage modulus is higher than the loss modulus, ensuring shape retention after extrusion. With the increase of shear stress, the loss modulus becomes higher than the storage modulus, ensuring a continuous flow through the nozzle as a pressure applied. This phenomenon confirms the practicability of the ink in DIW technology. Then the flexible MSC was prepared on the basis of interdigital electrode substrate and the BPVA-Pluronic electrolyte according to above. The digital photograph of as-prepared MSC is shown in Figure 6a. The active area of each single microelectrode was 7 mm × 1 mm and the average thickness of single microelectrode is 9 μm as depicted in Appendix A. The MSC shows the self-healing performance without external stimulation at room temperature. After 5 min of static state, the MSC, which is cut into two pieces, is restored to the original shape in structure (Figure 5b and Appendix A). According to Figure 5c and Figure 6b, the printed MWCNT is conformably coated onto the layered cellulosic paper substrate with high specific surface area, presenting wire-like nanostructures. The profile SEM image in Appendix A illustrates the random orientation and entangled coating behavior of carbon nanotubes on the surface of cellulose paper. On the meanwhile, as revealed in Figure 6d and Appendix A, the MWCNT close to the interface between the electrode and electrolyte are fully infiltrated and wrapped closely by hydrogel electrolyte, exhibiting a composite-like micromorphology. Due to the close interface contact, no adhesive is needed between interdigital microelectrode and substrate and the hydrogel electrolyte can fill carbon nanotube electrode well with strong interfacial strength. Such nanostructure not only enhances more activity for carbon nanotubes, which benefits the rapid electron-transfer and provides extra capacitance for energy storage devices, but also contributes to the flexibility and self-healing efficiency [38].

To estimate the electrochemical capability of the symmetric flexible MSC with BPVA-Pluronic acting as the electrolyte, the optimal operating voltage window of the fabricated MSC is chosen to be within 0~0.8 V according to the LSV test result shown in Figure 7a. The CV curves are presented with the scan rate from 5 mV s^−1^ to 100 mV s^−1^ in Figure 7b. All CV curves exhibit an approximately symmetrical rectangle, indicating ideal double-layer capacitance behaviors in process of charging and discharging (Figure 7c) [3,4]. At the current density of 20 μA cm^−2^, the as-prepared MSC shows the areal capacitance of 6.42 mF cm^−2^ and a volumetric capacitance of 801.9 mF cm^−3^. As the current density increases to 200 μA cm^−2^, the areal capacitance is still up to 5.56 mF cm^−2^ with the volumetric capacitance up to 694.6 mF cm^−3^. Capacity retention at 10 times current density is 86.60% and the voltage drop of discharge profiles is not observed obviously at high current density [44]. The potential-specific capacity curves also show a similar trend (Appendix A). The voltage increases/decreases linearly with the specific capacity after a brief platform during constant current charging/discharging process, showing a nearly with a constant slope value. With the increase of current density, the specific capacity decreases accordingly. During the self-discharge process (Figure 7d), it takes 270 s, 2620 s and 9520 s for the MSC to drop from the original state to 0.6 V, 0.4 V and 0.2 V respectively. As shown in Appendix A, the stage with discharge rate faster than −0.508 mV s^−1^ is determined by fast discharge of the compact layer. During the stage between −0.508 mV s^−1^ and −0.059 mV s^−1^, electric charge transfers from compact layer to diffusion layer. After the stage, discharge of diffusion layer is dominant. The behavior is consistent with Stern model [45]. The exceptional rate capability, cycle performance (Appendix A) and satisfied areal capacitance (Appendix A) are originated from high affinity between the interface of applied materials [29,31].

The interfacial stability of integrated configuration ensures the mechanical properties of the as-prepared MSC [9,32]. To investigate the electrochemical capacity under various bending angles from 0° to 90°, the CV performance and GCD measurement are tested with bending shape to demonstrate the flexible property of the assembled MSC. In comparison of the conditions shown in Figure 8a, the CV curves of the soft MSC under 0°, 30°, 45°, 60° and 90° bending degrees are close to overlapped and the GCD curves (Figure 8b) also remain the homologous shapes at various bent angles. As revealed in Appendix A, capacitance retention of the CV test is 97.59% from 0° to 90°, implying the excellent durability and mechanical robustness of the BPVA-Pluronic electrolyte-based MSC, as well as the excellent flexibility of the components which would not suffer severe decrease of the electrochemical performance under deformation [3].

Stability of electrochemical properties under multiple mechanical damages is the major challenge to develop durable MSC [46]. In order to overcome the defect of traditional substrate that self-healing ability is decreased by the compression deformation of the basement, cellulosic paper is selected as substrate of MSC [1,34]. During the physical damaged/healing process, the flexible MSC pieces cut in half are put to fitting closely for self-healing. Due to the effective contact of the electrode and electrolyte free from dislocation, the diol-borate ester bonding in the PVA-based network will regenerate at the interface induced by micellar network, thus leading to the healing of ion migration of BPVA-Pluronic electrolyte [3,39]. In addition, the interdigital microelectrodes reconnect between damaged interfaces by the hydrogen bond of MWCNT, resulting in the restoration of the electron transport [47,48,49]. As shown in Figure 8c,d, after a train of physical damaged/healing cycles at the same position, the flexible MSC can still reveal approximately rectangular CV curves and isosceles triangular GCD profiles, indicating ideal energy storage behavior of MSC [50]. Further, the specific capacitances of the self-healing MSC are investigated by CVs and GCDs. The special capacitance of the original MSC and the MSC after first healing are 5.76 mF cm^−2^ and 5.67 mF cm^−2^ according to the GCDs at the current density of 150 μA cm^−2^. The first and second GCD profiles nearly coincide, which suggests that the dense cellulose paper contributes to the optimal alignment of damaged interdigital microelectrodes and reduces the electrochemical performance degradation caused by misalignment during device recovery [35]. And after the 5th healing, the capacity retention ratios calculated from the CV curves and GCD profiles are about 90.43% and 89.98%, and the voltage drop almost remains unchanged, implying the excellent capacitance recovery of the healing MSC. (Appendix A) The proper decrease in capacitance can attribute to the loss of mechanical friction to the interface and irregular arrangement of carbon nanotubes during repeated cutting/healing cycles [38]. The self-healing ability of as-prepared MSC adopting cellulosic paper based interdigital electrode and BPVA-Pluronic electrolyte outperforms previous reported self-healing MSC [51,52,53,54], and the repeatable self-healing capability can achieve without external stimulation.

## 4. Conclusions

In summary, we are able to design a modern self-healing polyelectrolyte (BPVA-Pluronic) through interleaving the Pluronic micellar network into the B-PVA gel matrix. Under the synergistic effect of dynamic borate ester bond, interchain hydrogen bond and plasticization of micellar network, the hydrogel owns efficient self-healable behavior, self-standing performance and excellent stretchability up to 1535%. Based on the hydrogel electrolyte, the flexible paper-based MSC is assembled directly via DIW method without interfacial bonding agent. Benefited from hydrophilicity and large specific surface area of cellulosic paper, MSCs exhibit satisfied interface compatibility with high specific capacitance and cycling life. Under various bending deformation, MSCs demonstrate an excellent capacitance retention (97.59%), free from interface delamination. Moreover, in the absence of external environment stimulation, the broken MSCs show initial structural self-healing behavior in 5 min and electrochemical restoration in 1 h at room temperature, realizing 90.43% capacitance retention after five breaking/self-healing cycles. This work brings a novel prospect to the construction of self-healing hydrogel system in portable wearable electronics and provides the strategy for material design of multifunctional integrated MSCs.

## Figures and Tables

**Figure 1 materials-14-01852-f001:**
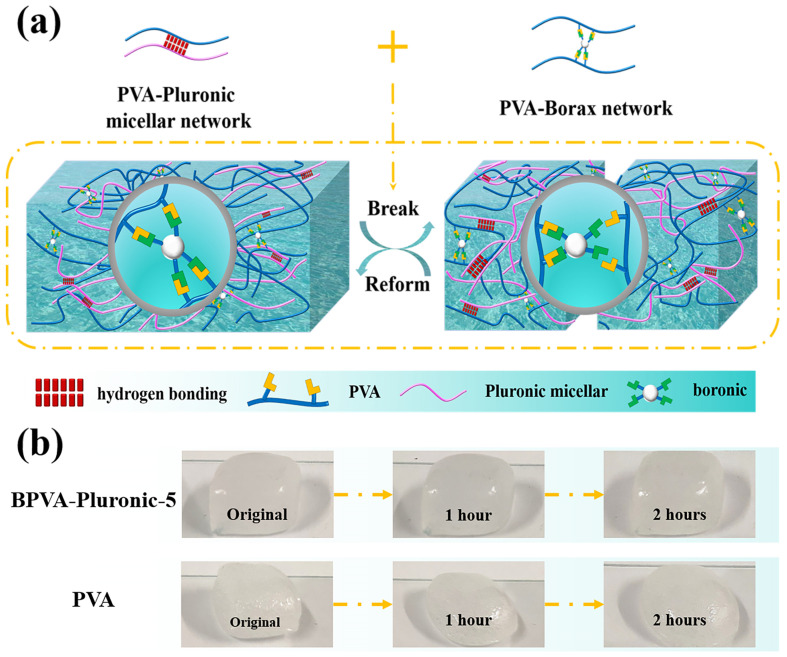
(**a**) Schematic illustration of the formation and structure of BPVA-Pluronic hydrogel. (**b**) Shape stability of BPVA-Pluronic-5 hydrogel and PVA hydrogel.

**Figure 2 materials-14-01852-f002:**
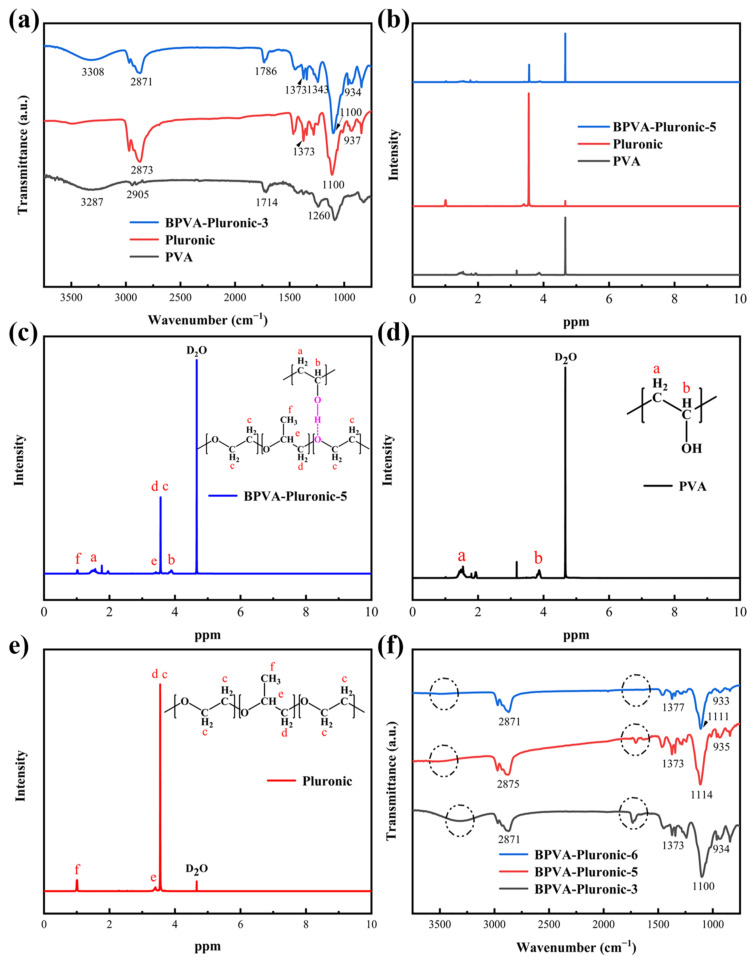
(**a**) The FTIR spectra of PVA, Pluronic and BPVA-Pluronic-3 hydrogel. (**b**) ^1^H NMR spectra of (**c**) BPVA-Pluronic-5 hydrogel, (**d**) PVA, (**e**) Pluronic. (**f**) The FTIR spectra of BPVA-Pluronic hydrogels with various weight ratio of PVA and Pluronic.

**Figure 3 materials-14-01852-f003:**
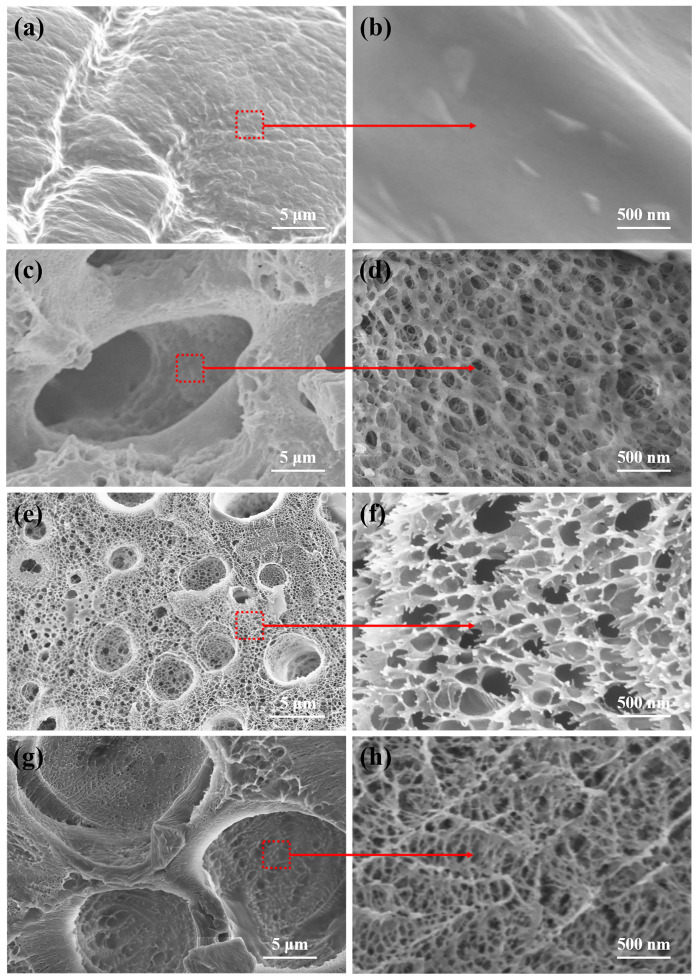
The cross-section SEM images of samples under different magnification: (**a**,**b**) pure PVA hydrogel, (**c**,**d**) BPVA-Pluronic-3 hydrogel, (**e**,**f**) BPVA-Pluronic-5 hydrogel, (**g**,**h**) BPVA-Pluronic-6 hydrogel.

**Figure 4 materials-14-01852-f004:**
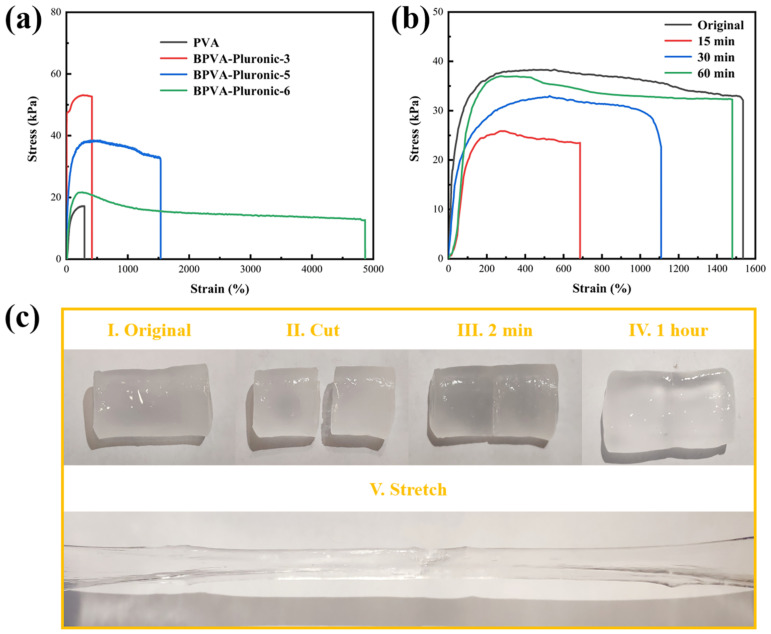
(**a**) Stress–strain curves of PVA hydrogel and BPVA-Pluronic hydrogels prepared from varied PVA/Pluronic ratio. (**b**) Stress–strain curves of the original BPVA-Pluronic-5 hydrogel and cut-healed BPVA-Pluronic-5 hydrogel for healing times of 15 min, 30 min and 60 min. (**c**) Digital photographs demonstrating the self-healing process of BPVA-Pluronic-5 hydrogel. I original state; II cut into two pieces; III self-healing in 2 min. IV self-healing in 1 h. V stretching after healing.

**Figure 5 materials-14-01852-f005:**
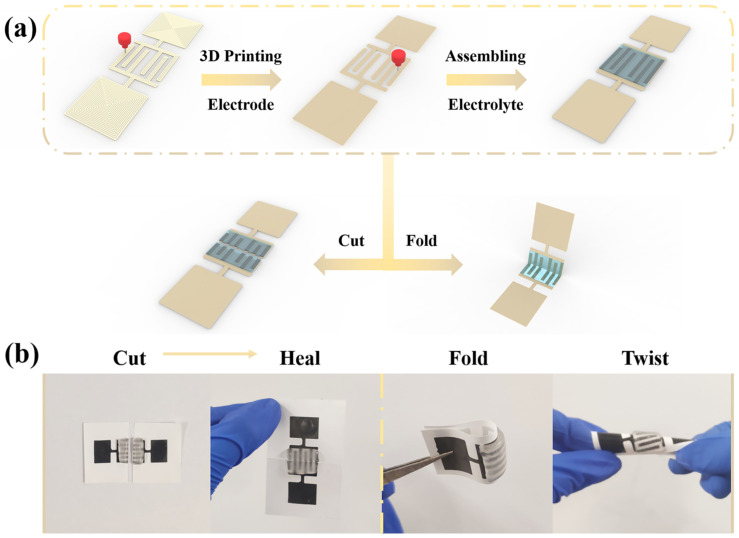
(**a**) Schematic illustrations of preparation process of the flexible MSCs. (**b**) Digital photographs of MSCs with damage/self-healing process, folding state and twisting state.

**Figure 6 materials-14-01852-f006:**
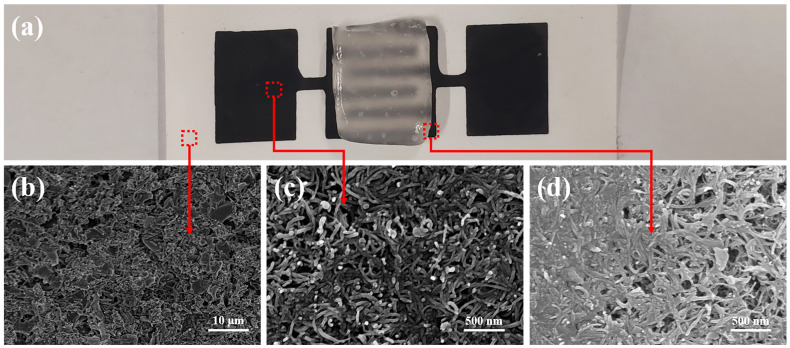
(**a**) Digital photograph of the as-assembled MSCs. (**b**) The surface SEM image of cellulose paper substrate. (**c**) The surface SEM image of MWCNT interdigitated microelectrodes. (**d**) The surface SEM image of the interface between BPVA-Pluronic hydrogel electrolyte and MWCNT interdigitated microelectrodes.

**Figure 7 materials-14-01852-f007:**
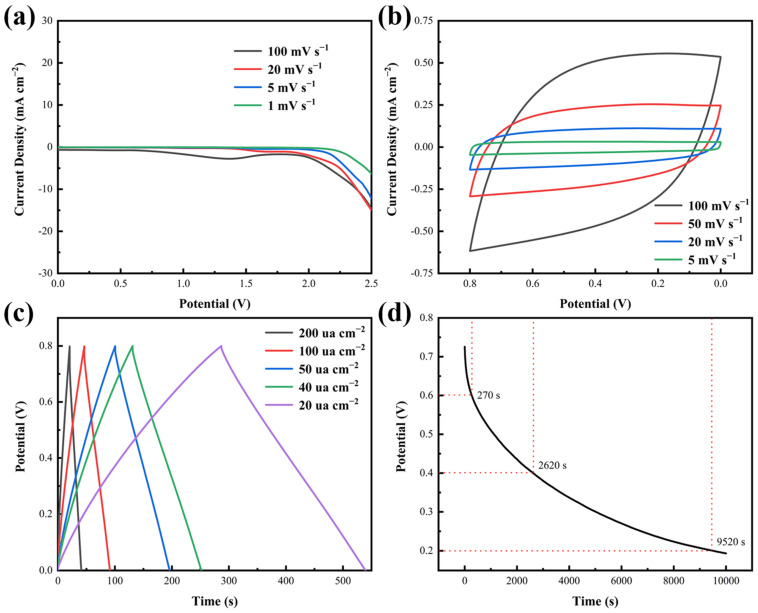
(**a**) LSV curves of the BPVA-Pluronic hydrogel electrolytes at various scan rates. (**b**) CV curves of the MSC at various scan rates. (**c**) GCD curves of the MSC at current densities. (**d**) Self-discharge curve of the MSC.

**Figure 8 materials-14-01852-f008:**
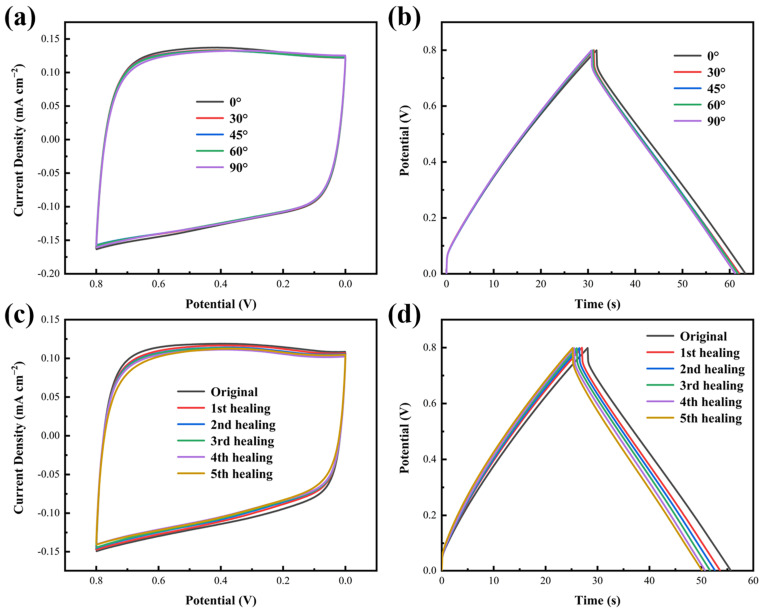
(**a**) CV curves (20 mV s^−1^) of the MSC at different bending angles. (**b**) GCD curves (150 μA cm^−2^) of the MSC at different bending angles. (**c**) CV curves (20 mV s^−1^) of the MSC at different damage/self-healing cycles. (**d**) GCD curves (150 μA cm^−2^) of the MSC at different damage/self-healing cycles.

## Data Availability

The data presented in this study are available on request from the corresponding author.

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
