# Peer review of "Self-Healing and Highly Stretchable Hydrogel for Interfacial Compatible Flexible Paper-Based Micro-Supercapacitor"

_materials, 2021, doi:10.3390/ma14081852_

Round 1

Reviewer 1 Report

No comments

Author Response

On behalf of my co-authors, thank you for your letter and for the reviewers’ comments concerning our manuscript entitled “Self-healing and Highly Stretchable Hydrogel for Interfacial Compatible Flexible Paper-based Micro-supercapacitor”. Those comments are all valuable and very helpful for revising and improving our paper, as well as the important guiding significance to our researches.

We have studied comments carefully, revised the manuscript in accordance with the reviewers’ comments and editorial notes, and carefully proof-read to minimize grammatical errors. In order to provide a convenient reading, the revised paragraphs are marked by Track Changes in red in the revised manuscript. The main corrections in the paper and the responses to the reviewers’ comments and editorial notes are as following:

Reviewer 1: No comments.

Authors’ Response: We greatly appreciate the respected reviewer for her or his time and kind suggestions. We have made the revisions according to your suggestions, and the manuscript has been checked to ensure flawless logic and language quality. Thanks again for your kind guidance and hard work.

Thank you and best regards,

Prof. Dr. Xinhua Xu

Reviewer 2 Report

Authors have presented an interesting manuscript. It would be very valuable addition to the literature. However I do have some points of concern. They are listed below:

  1. Did authors do elemental analysis as well? It would be valuable to add EDX data from SEM-EDX study.

  2. In lines 195-201, authors make a comparison of PVA/pluronic hydrogels with pure PVA hydrogels. However, I miss the data in Figure 4.

  3. Caption 4b needs to be more clear. Even though it is clear from the text, caption doesn’t explain the time-dependent healing process.

  4.  Did the authors perform similar time-dependent cut-healing process with pure PVA? That data should be added in the supplement for comparison purposes.

  5. Authors do not clearly explain about sample preparation for EIS studies in experimental methods. It need to be added.

  6. Figure 7a: caption is wrong. It represents Nyquist plot indicating real and imaginary impedance but not ionic conductivity directly. Of-course Ionic conductivity can be calculated from it. But the graph doesn’t represent that. Authors should extract and calculate the exact values and put them in a table for this study.

  7. Also, what was the impact of changing PVA/pluronic ratio on ionic conductivity? Since authors mention that “The well ion transport of BPVA-Pluronic electrolyte is mainly attributed to the following reasons. 1) The plasticizing effect of the Pluronic elastomer results in the viscosity reduction of hydrogel. 2) 3D network structure provides an efficient ion transport channel. 3) Properly increasing the concentration of KCl can produce generous of ions”. The ratio should have an impact.

  8. Authors mentioned in line 243 that “As Figure S2 shown, the rheological behavior of the carbon nanotube dispersion was estimated and the shear thinning phenomenon indicates the practicability of the ink in DIW technology.”
    Can they elaborate in the manuscript how it improves the practicality of application of this ink. Not every reader would be aware of the design specifications for the ink.

  9. Authors have indicated that average thickness of each microelectrode was 9 um and referred to figure S3. How was this figure generated? And it is not clearly depicted. Colormap is not uniform to see the height difference. Probably it is a result of data analysis. But I can not clearly see it in the figure. Also, what were the total number of micro-electrodes? From Figure S3, it seems that deposition was not very sharp and there was deposition in-between regions also?

  10. What was the thickness of the hydrogel layer on MSC?

  11. For figure 8, were the tests done in bended conditions or tests were done after bending and recovery of the MSC device at different angle. I miss the experimental explanation.

  12. Grammatical and English needs to improve. In introduction, some of the sentences were written in quite complex way and makes it difficult to read. I will not go through each one of them, but authors should carefully check or take help of editorial team from the journal.

Author Response

On behalf of my co-authors, thank you for your letter and for the reviewers’ comments concerning our manuscript entitled “Self-healing and Highly Stretchable Hydrogel for Interfacial Compatible Flexible Paper-based Micro-supercapacitor”. Those comments are all valuable and very helpful for revising and improving our paper, as well as the important guiding significance to our researches.

We have studied comments carefully, revised the manuscript in accordance with the reviewers’ comments and editorial notes, and carefully proof-read to minimize grammatical errors. In order to provide a convenient reading, the revised paragraphs are marked by Track Changes in red in the revised manuscript. The main corrections in the paper and the responses to the reviewers’ comments and editorial notes are as following:

Reviewer 2: 1. Did authors do elemental analysis as well? It would be valuable to add EDX data from SEM-EDX study.

Authors’ Response: We thank the referee for positive evaluation and useful suggestions. EDX data and related discussion have added in the revised text.

To demonstrate the elemental composition and distribution of hydrogel, EDX spectrums are present in Figure S1. The polymer composed of C and O elements forms the matrix of the crosslinking network and B elements are distributed in the network skeleton as crosslinking sites. The distribution of K and Cl elements in the hydrogel is uniform, indicating the sufficient infiltration of salt solution.

  1. In lines 195-201, authors make a comparison of PVA/pluronic hydrogels with pure PVA hydrogels. However, I miss the data in Figure 4.

Authors’ Response: Thanks for your comment and advice. In the former comparison, we directly cited the data of pure PVA hydrogel in reference. We have added the stress−strain curve of pure PVA hydrogel we made (the same preparation process without the addition of Pluronic) in Figure 4a. The fracture strength of the pure PVA hydrogel is 17.08 kPa and the fracture elongation is 294%. Accordingly, the description of the performance comparison has been modified.

  1. Caption 4b needs to be more clear. Even though it is clear from the text, caption doesn’t explain the time-dependent healing process.

Authors’ Response: Thanks for your kind suggestion. We have modified the text in detail in the corresponding position.

  1. Did the authors perform similar time-dependent cut-healing process with pure PVA? That data should be added in the supplement for comparison purposes.

Authors’ Response: Thank you very much for your comment. The stress−strain curves of the original pure PVA hydrogel and cut-healed pure PVA hydrogel for healing times of 6 h have been supplemented in Figure S3.

The pure PVA hydrogel was cut in half with scissors and subsequently the two cross-sections were put together into contact at room temperature in absence of external environment stimulation. After 1 h of self-healing, the fracture interfaces are connected together but the self-healed PVA hydrogel is too fragile to bear tensile test. After 6 h of self-healing, the fracture strain of the healed hydrogel is 246% with healing efficiency of 83.67%. The limited healing efficiency is mainly attributed to the reason that the initially formed dynamic borate ester bonds restrict the follow-up motion of molecular chains on the fracture surface.

  1. Authors do not clearly explain about sample preparation for EIS studies in experimental methods. It need to be added.

Authors’ Response: Thank you for your kind consideration. We are sorry for not addressing the experimental information clearly. We have added the suggested detailed experimental information of sample preparation for EIS studies in Electrochemical Characterization section.

The as-prepared hydrogel electrolyte was pressed between two slides to form the polymer film. The hydrogel film was cut in appropriate size of Φ16 mm*1.28 mm and then attached between two stainless steel sheets of Φ22 mm*0.5 mm. Both sides of test sample were connected to positive and negative poles of electrochemical workstation respectively.

  1. Figure 7a: caption is wrong. It represents Nyquist plot indicating real and imaginary impedance but not ionic conductivity directly. Of-course Ionic conductivity can be calculated from it. But the graph doesn’t represent that. Authors should extract and calculate the exact values and put them in a table for this study.

Authors’ Response: Thank you very much for your comment. We have modified the caption of Figure 7a. The exact values of the studied electrolytes (including the samples in Figure S4) have been supplemented in Table S2.

  1. Also, what was the impact of changing PVA/pluronic ratio on ionic conductivity? Since authors mention that “The well ion transport of BPVA-Pluronic electrolyte is mainly attributed to the following reasons. 1) The plasticizing effect of the Pluronic elastomer results in the viscosity reduction of hydrogel. 2) 3D network structure provides an efficient ion transport channel. 3) Properly increasing the concentration of KCl can produce generous of ions”. The ratio should have an impact.

Authors’ Response: Thanks for your concerned question. The frequency dependent impedance plots of the BPVA-Pluronic hydrogel electrolytes with varied PVA/Pluronic ratio have been supplemented in Figure S4.

With the increase of Pluronic mass ratio, BPVA-Pluronic-5 electrolyte has better ionic conductivity than BPVA-Pluronic-3 electrolyte, originating from the synergistic effect of plasticizing effect and efficient stereoscopic micro crosslinking network. However, there is no obvious change between ionic conductivity of BPVA-Pluronic-5 and BPVA-Pluronic-6. This phenomenon mainly attributes to the reason that the larger void diameters and plasticizing effect of BPVA-Pluronic-6 counteract each other.

  1. Authors mentioned in line 243 that “As Figure S2 shown, the rheological behavior of the carbon nanotube dispersion was estimated and the shear thinning phenomenon indicates the practicability of the ink in DIW technology.”

Can they elaborate in the manuscript how it improves the practicality of application of this ink. Not every reader would be aware of the design specifications for the ink.

Authors’ Response: Thank you very much for your suggestion. More discussions about rheological behavior of ink have been added in the revised manuscript.

As Original Figure S2 shown, the rheological behavior of the carbon nanotube dispersion was estimated and the shear thinning phenomenon indicates that the ink is non-Newtonian fluid. At the plateau range from 0.1 to 1 Pa, the storage modulus is higher than the loss modulus, ensuring shape retention after extrusion. With the increase of shear stress, the loss modulus becomes higher than the storage modulus, ensuring a continuous flow through the nozzle as a pressure applied. This phenomenon confirms the practicability of the ink in DIW technology.

  1. Authors have indicated that average thickness of each microelectrode was 9 um and referred to figure S3. How was this figure generated? And it is not clearly depicted. Colormap is not uniform to see the height difference. Probably it is a result of data analysis. But I can not clearly see it in the figure. Also, what were the total number of micro-electrodes? From Figure S3, it seems that deposition was not very sharp and there was deposition in-between regions also?

Authors’ Response: Thanks for your concerned question.

Authors have indicated that average thickness of each microelectrode was 9 um and referred to figure S3. How was this figure generated? And it is not clearly depicted. Colormap is not uniform to see the height difference. Probably it is a result of data analysis. But I can not clearly see it in the figure.

Original Figure S3a is the 3D structure of the interface between the edge of MWCNT interdigitated microelectrodes and cellulose paper substrate. Original Figure S3b is the 3D structure of a single MWCNT interdigitated microelectrode. Different colors represent the relative height in the same picture. As shown in the figure below (Please see the attachment.):

During the process of interdigital microelectrode, the movement track of extrusion syringe is shown as the figure below (Please see the attachment.):

The red area of Original Figure S3 corresponds to the movement track, thus accumulating thicker electrode materials. The green and blue area of Original Figure S3b is the middle of single interdigital microelectrode with less electrode materials. Local height difference is form by local fluidity of electrode ink. According to the relative height value in the figure below (Please see the attachment.):

We can approximately fit the cross section of single interdigital microelectrode to the model below (Please see the attachment.):

Therefore, the average thickness of each microelectrode is nearly 9 μm.

Also, what were the total number of micro-electrodes?

According to the digital photo above, the total number of interdigital microelectrodes is 6. Each side has 3 interdigitations respectively.

From Figure S3, it seems that deposition was not very sharp and there was deposition in-between regions also?

The green area in Original Figure S3a is mainly attributed to height difference caused by surface roughness of cellulose paper and error in measurement process. The red and yellow points of in-between regions is owing to sampling error of S neox Five Axis.

  1. What was the thickness of the hydrogel layer on MSC?

Authors’ Response: Thanks for your concern. The thickness of the hydrogel layer on MSC is 1.5 mm.

  1. For figure 8, were the tests done in bended conditions or tests were done after bending and recovery of the MSC device at different angle. I miss the experimental explanation.

Authors’ Response: Thanks for pointing this out. The tests were done in bended conditions. We have revised the expression of the original text to avoid ambiguity.

  1. Grammatical and English needs to improve. In introduction, some of the sentences were written in quite complex way and makes it difficult to read. I will not go through each one of them, but authors should carefully check or take help of editorial team from the journal.

Authors’ Response: Thanks for kind reminder. Per suggestion, we have very carefully made revisions in introduction, optimized the structure of some complex expressions and tried to avoid any grammar or syntax error. The detailed revisions and depictions have been highlighted with Track Changes in red in the revised manuscript.

Thank you and best regards,

Prof. Dr. Xinhua Xu

Reviewer 3 Report

Firstly - very good piece of work with a lot of really nice measurements and clear explanations. 

Changes suggested here reflect only very minor edits required for publication.

Line 17 add PEO and PPO here as this is the first time these are referenced

Line 35 change "are suffered" to suffer

Line 48 delete duplicate "and type"

Line 105 change "controlling" to controlled

Line 107 you mention its a vacuum oven, readers might be interested to know approx pressure

Line 110 Consider changing to "Structural Characterization" to distinguish from other types

Line 161 (twice) change "FT-IR" to FTIR to be consistent with the text

Line 234 replace "well" to good

Line 236 remove word "Properly"

Line 237 replace word "generous" with surplus

Line 341 remove "Please add:"

References - there are 54 references listed but I don't find all of these in the paper. Please check this and make sure that these are cited or removed.

Two main overall comments:

1) Lots of references to supplementary information, consider including some more in the paper if its part of the narrative.

2) The title states this is a supercapacitor, consider including supercapacitor characteristic measurements such as self-discharge rate & charge profile to show the key metrics.

Again, really good paper - well done.

Author Response

On behalf of my co-authors, thank you for your letter and for the reviewers’ comments concerning our manuscript entitled “Self-healing and Highly Stretchable Hydrogel for Interfacial Compatible Flexible Paper-based Micro-supercapacitor”. Those comments are all valuable and very helpful for revising and improving our paper, as well as the important guiding significance to our researches.

We have studied comments carefully, revised the manuscript in accordance with the reviewers’ comments and editorial notes, and carefully proof-read to minimize grammatical errors. In order to provide a convenient reading, the revised paragraphs are marked by Track Changes in red in the revised manuscript. The main corrections in the paper and the responses to the reviewers’ comments and editorial notes are as following:

Reviewer 3: Firstly - very good piece of work with a lot of really nice measurements and clear explanations.

Changes suggested here reflect only very minor edits required for publication.

Authors’ Response: We would like to thank the reviewer for the deep and thorough reviewing of our manuscript, and to enriching us with their positive and valuable comments which helped us to improve the quality of the manuscript. Thanks again for your kind guidance and hard work.

  1. Line 17 add PEO and PPO here as this is the first time these are referenced.

Authors’ Response: Thanks for pointing this out. We have added the note in the abstract these are referenced for the first time.

  1. Line 35 change "are suffered" to suffer.

Authors’ Response: Thank you very much for your suggestion. We have changed "are suffered" to "suffer" in the corresponding paragraph.

  1. Line 48 delete duplicate "and type".

Authors’ Response: Thank you very much for your consideration. We have deleted duplicate "and type" in the corresponding paragraph.

  1. Line 105 change "controlling" to controlled.

Authors’ Response: Thanks for your suggestion. We have changed "controlling" to "controlled" in the corresponding paragraph.

  1. Line 107 you mention its a vacuum oven, readers might be interested to know approx pressure.

Authors’ Response: Thanks for your kind suggestion. The absolute pressure in the vacuum drying oven is 15.20 kPa. We have added the note in Fabrication of flexible electrochemical MSC section.

  1. Line 110 Consider changing to "Structural Characterization" to distinguish from other types.

Authors’ Response: Thanks for your advice. We have changed "Characterization" to "Structural Characterization" to distinguish from other characterization.

  1. Line 161 (twice) change "FT-IR" to FTIR to be consistent with the text.

Authors’ Response: Thank you very much for your suggestion. We have changed "FT-IR" to "FTIR" in two corresponding positions.

  1. Line 234 replace "well" to good.

Authors’ Response: Thanks for your suggestion. We have changed "well" to "good" in the corresponding paragraph.

  1. Line 236 remove word "Properly".

Authors’ Response: Thanks for pointing this out. We have removed "Properly" in the corresponding paragraph.

  1. Line 237 replace word "generous" with surplus.

Authors’ Response: Thank you very much for your consideration. We have replaced word "generous" with "surplus" in the corresponding paragraph.

  1. Line 341 remove "Please add: ".

Authors’ Response: Thanks for your advice. We have removed "Please add: " in the corresponding paragraph.

  1. References - there are 54 references listed but I don't find all of these in the paper. Please check this and make sure that these are cited or removed.

Authors’ Response: Thanks for your concerned question. The references of [44] to [54] are listed in Table S3 in Supplementary Materials. According to your kind advice below, I have cited references in their related fields in the manuscript.

  1. Two main overall comments:

1) Lots of references to supplementary information, consider including some more in the paper if its part of the narrative.

Authors’ Response: Thanks for your kind suggestion. We evaluated the work of references in supplementary materials again and found a lot of excellent points. All the references have been cited in manuscript and a reference with low correlation has been removed from Table S3.

  1. 2) The title states this is a supercapacitor, consider including supercapacitor characteristic measurements such as self-discharge rate & charge profile to show the key metrics.

Again, really good paper - well done.

Authors’ Response: Thank you very much for your comment and advice.

The corresponding potential-specific capacity curves of the MSC have been supplemented in Figure S8. The voltage increases/decreases linearly with the specific capacity after a brief platform during constant current charging/discharging process, showing a nearly with a constant slope value. With the increase of current density, the specific capacity decreases accordingly, showing a trend similar to the GCD curves (Figure 7d).

The corresponding self-discharge and self-discharge rate curves of the MSC have been supplemented in Figure S9. During the self-discharge process (Figure S9a), it takes 270s, 2620s and 9520s for the MSC to drop from the original state to 0.6V, 0.4V and 0.2V respectively, which is consistent with Stern model. As shown in Figure S9b, the stage with discharge rate faster than -0.508 mV s-1 is determined by fast discharge of the compact layer. During the stage between -0.508 mV s-1 and -0.059 mV s-1, electric charge transfers from compact layer to diffusion layer. After the stage, discharge of diffusion layer is dominant.

Thank you and best regards,

Prof. Dr. Xinhua Xu

Round 2

Reviewer 2 Report

I am satisfied with the reply provided by the authors. Also, I once again appreciate the detailed work which has been done in this manuscript.